# Voriconazole Pharmacokinetics Administered at 4 mg/kg IM and IV in Nursehound Sharks (*Scyliorhinus stellaris*) Under Human Care

**DOI:** 10.3390/vetsci12010017

**Published:** 2025-01-03

**Authors:** Daniela Cañizares-Cooz, Daniel García-Párraga, Sonia Rubio-Langre, Teresa Encinas, Pablo Morón-Elorza

**Affiliations:** 1Department of Pharmacology and Toxicology, Faculty of Veterinary Medicine, Complutense University of Madrid, Av. Puerta de Hierro s/n, 28040 Madrid, Spain; sonrubio@ucm.es (S.R.-L.); tencinas@vet.ucm.es (T.E.); p-moron@hotmail.com (P.M.-E.); 2Fundación Oceanogràfic de la Comunitat Valenciana, C/Eduardo Primo Yúfera (Científic), 1B, 46013 Valencia, Spain; dgarcia@oceanografic.org; 3Veterinary Services, Oceanogràfic, Ciudad de las Artes y las Ciencias, C/Eduardo Primo Yúfera (Científic), 1B, 46013 Valencia, Spain

**Keywords:** sharks, mycosis, voriconazole, pharmacology

## Abstract

Sharks and rays are a vital part of the collections of many aquariums around the world. The study of these species kept under human care has revealed a number of health conditions, highlighting the necessity of the further development of treatment plans. Although fungal infections are uncommon in elasmobranchs, they can lead to high mortality rates in aquariums. Some promising results have been reported with voriconazole, but pharmacokinetic studies are needed. In this study, voriconazole was administered IV and IM to six Nursehound sharks (*Scyliorhinus stellaris*) at a dose of 4 mg/kg, with blood samples subsequently collected for analysis. Pharmacokinetic analysis was carried out, and results showed a mean peak plasma concentration (Cmax) ± SEM of 3.00 ± 0.23 µg/mL following IM administration. Terminal half-lives for IV and IM injections were 7.94 ± 0.49 h and 4.27 ± 1.04 h, respectively. These results are promising; however, further susceptibility testing in elasmobranch fungal infections is necessary to evaluate the effectiveness of the posology proposed in this study.

## 1. Introduction

Elasmobranchs, which include sharks, rays, and skates, are frequently kept in aquariums worldwide. The maintenance of these species under human care has facilitated the study and understanding of their diseases, as well as the development of treatments. In 2013, a review of the pathologies affecting elasmobranchs in aquariums was published, showing that fungal diseases represent 0.6% of all the diseases registered in the database consulted [1]. Despite this low incidence, fungal infections are associated with high mortality rates in elasmobranchs and can severely affect their health. Common fungal pathogens involved in these cases are *Fusarium* spp., *Paecilomyces* spp., *Exophiala* spp., and *Mucor* spp., with *Fusarium* spp. being the most frequently isolated [2]. Fusariosis has also been reported in other aquatic species such as teleost fish, sea turtles, and marine mammals, both in aquariums and in the wild [3].

Most cases of fusariosis in elasmobranchs have been reported in the *Sphyrna* genus, commonly referred to as “bonnethead or hammerhead shark disease”. Fungal diseases typically cause ulcerative skin lesions, especially on the head, fins, and lateral line [4,5]. In advanced stages, the infection can progress and become systemic, affecting the liver, spleen, kidneys, celomic cavity, and the brain, ultimately leading to death. Due to immunosuppression, opportunistic bacterial infections commonly develop, further complicating the animal’s condition [6,7].

The therapeutic management of these infections is challenging. Initially, reported treatments included a combination of antibiotics (amikacin, ceftazidime, chloramphenicol), antifungal drugs (terbinafine, itraconazole) and anti-inflammatories (prednisolone). However, these treatments did not improve the clinical signs of the affected animals [8,9]. Azole antifungal drugs have been used in several cases of fungal diseases, particularly fusariosis. Itraconazole has been employed with poor outcomes in most cases. However, voriconazole prescription has produced remission of clinical signs in some animals, demonstrating a low minimum inhibitory concentration (MIC) against *F. solani* (1 to 2 µg/mL) in vitro, although susceptibility testing in elasmobranch mycoses reported a higher MIC for *F. solani* (8 µg/mL) [10,11]. Anecdotal evidence indicates that clinical outcomes do not consistently correlate with *F. solani* MICs, with reports of successful fusariosis treatment in elasmobranchs when voriconazole plasma concentrations of 1.2–1.65 µg/mL were achieved [12]. The main administration routes are topical and oral, with doses ranging from 3 mg/kg to 50 mg/kg, though the optimal dosing and the voriconazole therapeutic range in elasmobranchs remains unclear due to the lack of PK data [11,12].

The effectiveness of azole antifungals in elasmobranchs is uncertain, as most treatment protocols are based on studies conducted in teleosts. The empirical administration of azole antifungal drugs has been reported in fish, with a wide range of doses (10–50 mg/kg) [4]. Itraconazole has been administered to various species at doses of 1–5 mg/kg p.o every 24 h, ketoconazole at 2.5–10 mg/kg p.o; i.m and i.c for 10 days, similar to miconazole, which has been prescribed at 10–20 mg/kg via the same routes, every 24 h [13]. Voriconazole has been tested in exotic species such as belugas (*Delphinapterus leucas leucas*) at the dose of 3 mg/kg p.o [14] and various elasmobranchs, including the bonnethead shark (*Sphyrna tiburo*) at 4–6 mg/kg p.o and 4 mg/kg i.v, and 30 mg/kg p.o, s.i.d; scalloped hammerhead shark (*Sphyrna lewini*) at doses ranging from 12 mg/kg p.o to 50 mg/kg p.o; roughtail stingray (*Dasyatis centroura*) at 3–4 mg/kg p.o, s.id; and lesser devil ray (*Mobula hypostoma*) at 4 mg/kg i.m q48h [11,12,15,16].

Voriconazole PK has been studied in several species including domestic animals like dogs and cats, using very similar doses (4–6 mg/kg p.o and i.v) [17,18], as well as exotic species such as African penguins (*Spheniscus demersus*) at 5 mg/kg p.o [19], green sea turtle (*Chelonia mydas*) at 10 mg/kg p.o [20], red-eared sliders (*Trachemys scripta elegans*) at 10 mg/kg s.c [21], and western pond turtles (*Actinemys marmorata*) at 10 mg/kg s.c [22]. In elasmobranchs, voriconazole PK has been studied only in the undulate skate (*Raja undulata*) at 4 mg/kg i.v and i.m [23].

The physiological differences between teleosts and elasmobranchs make extrapolating dosing regimens challenging. As a result, PK studies in cartilaginous fish are essential. Recent articles on anti-inflammatory drugs’ PK in elasmobranchs have highlighted interspecies differences in drug metabolism and pharmacokinetics, as well as differences when compared to teleosts [24,25,26].

Given the impact of fungal diseases on elasmobranchs and the limited information available on antifungal treatment for this group of animals, further PK studies are necessary. Voriconazole has shown promising outcomes in reported cases, albeit with a wide range of doses (from 4 mg/kg to 50 mg/kg), making it unclear which is the most effective and appropriate dose. In this study, we aimed to evaluate the PK properties of voriconazole administered IV and IM in nursehound sharks. This research intends to provide insights into better treatment definitions of fungal diseases in these species, providing clinicians working with sharks with a scientific basis that can help prescribe safe and effective pharmacotherapeutic treatments.

## 2. Materials and Methods

### 2.1. Animals and Environment

The Generalitat Valenciana (2024-VSC-PEA-0091) authorized the experimental protocol in accordance with the Animal Care and Welfare Committee at the Oceanogràfic of Valencia under the project code OCE-22-19.

For this study, we included a group of six adult nursehound sharks (*Scyliorhinus stellaris*) that had been kept for the last 5 years in the facility of the Oceanogràfic of Valencia.

A protocol was established for the eligibility of the animals for the study, and routine blood test analysis including hematology and biochemistry were performed prior to the study. All the animals were determined to be clinically healthy based on a detailed veterinary examination [27].

Sharks were located in a 10,000 L cylindrical tank in the quarantine facilities of the aquarium. Each individual was identified based on unique spot patterns, fin markings, and other distinctive characteristics. Environmental parameters were carefully controlled during the study. Water temperature was maintained at 18 °C, salinity was 34 g/L, and pH levels ranged between 7.9 and 8.1. The tanks were provided with shaded areas and structures that allowed hiding and protection as well as environmental enrichment to ensure animal welfare. Animals were fed with cephalopods and teleosts once a day, six days per week, and 24 h was established as the minimum fasting period before the study. All sharks were weighed using a crane scale (GRAM CR 150-S, Gram Precision S.L., l’Hospitalet de Llobregat 08907, Barcelona, Spain). The weights of the animals ranged from 2.00 to 3.68 kg, with a mean weight of 2.94 ± 0.58 kg standard deviation (SD). All animals were considered adults based on the maturity size ranges reported for the species, and the presence or absence of claspers was used for sexual classification [28].

### 2.2. Experimental Design

The present study was designed as a prospective experimental trial. During sampling, animals were closely monitored to identify any clinical signs or behavioral changes that could be related to blood collection, handling, or adverse drug effects. All subjects received the same dose of 4 mg/kg voriconazole (Voriconazole Normon^®^ 200 mg powder for injection, Barcelona 08013, Spain), diluting each vial in 10 mL of sterile water for injection. Administration was first intravenous (IV) and then intramuscular (IM) with a minimum eight-week washout period between each study route.

A rubber net was used for the capture of the sharks, and they were manually restrained underwater in a dorsoventral position for the IV administration. Intravenous injection was performed using a lateral approach to the caudal vasculature (Figure 1A) at a slow and constant rate, assuring that vascular access was maintained throughout the administration. Voriconazole was administered using a 23-gauge needle and the mean volume given ± SD was 0.68 ± 0.11 mL.

For IM administration, animals were positioned in ventral recumbency, with most of their body submerged. Injections were given dorsally into the epaxial musculature, in the marginal region to the dorsal fin (Figure 1B). Pressure was applied for a few seconds to the injection site after administration to minimize potential drug leakage [29].

Blood extraction was performed via a lateral approach to the caudal vessels using a 23-gauge needle attached to a 1 mL syringe. The sharks were placed in dorsal recumbency inducing tonic immobility. The tail and caudal half of the animal were kept above the water’s surface; the gills and head remained submerged to allow for continuous respiration. A volume of 0.4 mL blood was taken from each animal before drug administration and at previously determined intervals, following voriconazole IV or IM administration: 15, 30 min and 1, 1.5, 2, 4, 8, 12, 24, and 36 h. The sampling time points in our study were strategically chosen based on preliminary PK data of voriconazole in elasmobranchs, specifically in undulate skates and nursehound sharks [9]. This approach was further supported by an extensive literature review and a detailed examination of previously published studies on a wide range of animal species, with an emphasis on elasmobranchs [18,21,23,30,31,32,33]. Total handling time for each blood sampling including animal capture and blood collection was always kept under 2 min.

### 2.3. Blood Processing

Following each extraction, blood was transferred into 1 mL lithium heparin tubes. Aquisel^®^ (AQUISEL S.L., Abrera 08630, Spain). Samples were kept at 4 °C and processed in the aquarium’s laboratory within 45 min of collection. Blood was centrifuged at 590 G for five minutes using an Ortoalresa^®^ centrifuge (Ortoalresa^®^ RT106, Ortoalresa-Alvarez Redondo S.A., Daganzo de Arriba 28814, Spain). Plasma was separated and transferred to 1.5 mL Eppendorf tubes. The samples were frozen at −20 °C until voriconazole concentration analysis.

### 2.4. Voriconazole Quantification

The concentration of voriconazole was determined in each plasma sample using a reverse-phase high-performance liquid chromatographic (HPLC) method adapted from those previously described [34,35] and validated at the at the laboratory of the Department of Pharmacology and Toxicology of the Complutense University of Madrid for its use in elasmobranch plasma. A C18 column (Mediterranean Sea C-18 column; Teknokroma Analítica S.A., Barcelona 08173, Spain) was equipped to the HPLC system (Hitachi High-Tech Corporation; equipped with a 5160-model injection pump with manual purge valve, 5280-model auto-injector with cooling unit, and a 100 µL syringe, 5410-model variable ultraviolet detector, 08210, Barbera Del Valles, Barcelona, Spain). A solution of Acetonitrile and Milli-Q^®^ filtered H_2_O (60:40 [vol:vol]) was used as the mobile phase, which flowed at a rate of 0.8 mL/min. A wavelength of 262 nm was selected for the UV detector.

First, a volume of 100 µL of shark plasma was mixed with 25 µL HClO4 (1 M); then, the mixture was vortexed for one minute. Next, a volume of 400 µL methanol was added to the mixture, vortexed for 30 s, and then centrifuged at 2190 G for 5 min. Subsequently, the supernatant was extracted, resulting in 200 µL volume that was added to vials for the HPLC, 20 µL of which being injected by the machine into the chromatography system. A methanol solution of known voriconazole concentration was used to construct a calibration curve (ranging from 0.05 to 15 µg/mL), exhibiting linear absorbance at the studied concentrations (R^2^ > 0.99). The limit of the quantification resulted in 0.09 μg/mL, and inter- and intra-assay variabilities were always under 6%. Voriconazole mean recovery rate in nursehound shark plasma was determined by adding known concentrations of the drug to blank nursehound shark plasma, resulting in 80% (Sigma-Aldrich Química SA, Tres Cantos 28760, Madrid, Spain R).

### 2.5. Data Analysis

The maximum concentration in plasma (C_max_) and the time to reach it (T_max_) were directly determined from the plasma concentration vs. time data. Commercially available software was used for data analysis (PK Solutions^®^, version 2.0, Summit Research Services, Montrose, CO, USA) and the PK study was executed using a non-compartmental analysis model. The following PK parameters were determined, including elimination half-life (t_1/2β_), area under the plasma concentration–time curve to the last sampling time (AUC_t_), area under the plasma concentration–time curve extrapolated to infinity (AUC_inf_), volume of distribution (Vd), clearance (Cl), and mean residency time (MRT). The bioavailability (F) was calculated for each individual shark by dividing AUC_im_/AUC_iv_.

In this study, all plasma concentrations and estimated pharmacokinetic parameters are reported as their corresponding means ± standard error of the mean (±SEM). PK parameters for IV and IM administration were compared using a Wilcoxon signed-rank test.

## 3. Results

Voriconazole administered IV and IM was well tolerated by all animals; no adverse effects nor clinical abnormalities were observed during the study or four weeks after. Voriconazole mean plasma concentrations following 4 mg/kg IV and IM administrations are represented in Figure 2. Estimated PK parameters of voriconazole for both administration routes are shown in Table 1.

Voriconazole showed a fast absorption following IM administration with a T_max_ of 1.08 ± 0.15 h, and the bioavailability (F) was close to 50% (47.31 ± 8.69%). Plasma concentrations remained above the MIC50 threshold of 1 µg/mL for 6 and 10 h, and above the MIC90 threshold of 2 µg/mL for 2 and 4 h, following IM and IV administrations, respectively.

After IV administration, two distinct elimination phases were observed: an initial rapid phase lasting up to 4 h, followed by a prolonged and slower phase extending to 24 h. Both elimination phases can also be observed after IM administration. The volume of distribution was similar between the two routes, with 1.39 ± 0.09 L/kg for IV and 1.50 ± 0.118 L/kg for IM administration, showing no statistically significant difference. Clearance was faster following IM administration (0.29 ± 0.05 mL/min) compared to IV administration (0.12 ± 0.01 mL/min).

## 4. Discussion

The treatment of fungal diseases in sharks and rays is challenging due to the lack of evidence regarding dosage, administration routes, and treatment protocols in these species. In this situation, some authors recommend therapeutic drug monitoring (TDM) through plasma concentration measurement [12]. Pharmacokinetic studies provide a scientific foundation for the medical treatment of these diseases in elasmobranchs.

Azoles are one of the most frequently prescribed groups of antifungals for the treatment of disseminated mycosis in small animals and exotics and are widely used in avian and reptiles, primarily via oral administration, though parenteral routes have also been tested in some reptile and avian species [36,37]. However, information on the administration of antifungal drugs in elasmobranchs is sparse, and this PK study is one of the few available that involve the parenteral administration (IV and IM) of azole in fish species [23]. The parenteral administration of azole antifungals warrants further evaluation, as IV and IM administration routes have shown promising PK results with other drugs in elasmobranchs [25,26].

Voriconazole absorption was slightly faster in nursehound sharks (T_max_ = 1.08 h) when compared with other elasmobranch species such as the undulate skate (T_max_ = 1.33 h). Voriconazole mean peak plasma concentrations after 4 mg/kg IM administration produced comparable results in sharks (3.00 ± 0.37 μg/mL) and skates (2.98 ± 0.28 μg/mL) [23], and was higher than those values observed after the oral administration of 30 mg/kg s.i.d in bonnethead sharks (*Sphyrna tiburo*) (1.65 μg/mL) and 50 mg/kg b.i.d in hammerhead sharks (*Sphyrna lewini*) (1.2 μg/mL); these differences can be related to lower absorption or to the hepatic first pass effect following oral administration; however, further studies are required to explain the cause of these variations [12]. In birds, the absorption after oral administration is variable between species, and voriconazole plasma concentrations in mallard ducks (*Anas platyrhynchos*) after 10 mg/kg p.o reached 6.9 µg/mL [31], and in African grey parrots (*Psittacus erithacus timneh*), 5.68 µg/mL after 18 mg/kg p.o [38]. There is one study in three different species of falcons in which 12.5 mg/kg of voriconazole was administered intramuscularly and maximum plasma concentration levels were between 5.06 and 5.79 µg/mL [32]. In reptiles, the maximum plasma concentration after a multiple-dose regime of 10 mg/kg s.c, b.i.d of voriconazole was between 7.58 μg/mL in red-eared slider turtles (*Trachemys scripta elegans*) and 12.4 μg/mL in western pond turtles (*Actinemys marmorata*) [21,22]. Comparing the data available in other species, voriconazole C_max_ in nursehound sharks (3.00 ± 0.37 μg/mL) was lower than those reported in avian and reptile species, highlighting that in all the cases, the doses administered were significantly higher than those used in this study.

Plasma concentrations were above the MIC50 values reported for *Fusarium solani* during in vitro studies for a short period of time (6 to 10 h) [10]. MIC values of *Fusarium solani* isolated from elasmobranchs have been reported in one study (8 µg/mL) [11], which exceeds the maximum plasma concentration (3.00 ± 0.23) achieved after the IM administration of 4 mg/kg of voriconazole. This suggests that a higher dose could be required to effectively treat fusariosis. However, *Fusarium* has great variability regarding in vitro antifungal susceptibility to different drugs including voriconazole (1–8 μg/mL), and it is crucial to emphasize that in vitro MIC values do not always directly correlate with in vivo efficacy of antifungal treatments [39]. Furthermore, MIC values are not always correlated with clinical outcomes when compared directly with plasma concentration levels; other parameters such as the AUC/MIC ratio have been proposed as a more accurate value to predict efficacy, and unfortunately, more research is needed in this field [40]. In this sense, it is necessary to mention that published clinical reports of successful fusariosis treatment in elasmobranchs recorded an improvement in the disease when voriconazole plasma concentrations of 1.2–1.65 μg/mL were achieved [12]. Further in vitro and in vivo studies are needed to determine the therapeutic range for voriconazole in elasmobranch.

Mean bioavailability after IM administration was 47.31 ± 8.69%, which is similar to the bioavailability values reported after IM administration of voriconazole and other drugs in teleosts (51% after 1 mg/kg IM administration of meloxicam in Nile tilapias (*Oreochromis niloticus*) and elasmobranch species (53% in nursehound sharks after 1 mg/kg IM administration of meloxicam) [23,25,26,41]. However, this value is relatively low when compared with other mammalian species, such as humans (>90%), cats (>00%), and horses (>100%) [18,30,42]. Clinicians and researchers should note that voriconazole bioavailability has shown important variations in humans when administered orally with food, being reduced from 90% to 22% [43]. Data available in other species are limited: in mallard ducks (*Anas platyrhynchos*), the voriconazole bioavailability after 10 mg/kg p.o administration was 60.7%; in chickens it was 20% after 10 mg/kg p.o administration, which shows a great variability between these two avian species [31,33]. Drug leakage after IM administration is common due to the combination of skin and muscle characteristics in fish (including elasmobranchs); therefore, the bioavailability value obtained in this study falls within the range expected for sharks and rays [29]. Drug leakage could be reduced by using more concentrated formulations to decrease the administered volume, distributing the same volume in different areas of the body, or by sealing the injection point after administration. These techniques could be suggested to improve the bioavailability of drugs administered intramuscularly in elasmobranchs, and further studies are needed. Improving drug bioavailability was not the objective of this study, as we aimed to evaluate the current administration protocols for voriconazole in clinical management, and additional PK studies are needed to test the aforementioned hypotheses in elasmobranchs.

Voriconazole t_1/2β_ in nursehound sharks was significantly longer after IV administration (7.94 ± 0.49 h) compared to IM administration (4.27 ± 1.04 h). Both values were shorter than those observed in undulate skates after the administration of the same dose of the drug (11.18 ± 1.32 h for IV and 9.59 ± 1.38 h for IM administration). This parameter can vary significantly between species; for instance, the voriconazole elimination half-life rate after IV administration in mammals such as cats (12.43 ± 4.68 h) or horses (8.89 ± 2.31 h) is considerably different [18,30]. While there are no available data on the IM administration of voriconazole in mammals, studies in avian species have provided rather short t_1/2β_ values after IM administration in falcons (5.59 ± 2.63 h) [32].

The mean residency time (MRT) was shorter in the nursehound shark following both IV and IM administration (10.42 ± 0.57 and 6.43 ± 1.46 h, respectively) compared to other elasmobranch species such as undulate skates (13.17 ± 2.09 and 14.43 ± 2.19 h, respectively). MRT and t_1/2β_ values follow a similar pattern in nursehound sharks: both are shorter after IM than after IV administration. This suggests that in this species, elimination half-life might be more closely related to elimination processes, such as metabolism or excretion, rather than to distribution processes, in contrast to the undulate skates [23]. This observation is consistent with the fact that there are no statistically significant differences in Vd values between IV and IM routes (1.39 ± 0.09 and 1.50 ± 0.18 mL/min, respectively; *p* = 0.631), but differences have been found in clearance (Cl) values (0.12 ± 0.01 and 0.29 ± 0.05 mL/min, respectively; *p* = 0.006) in nursehound sharks.

Clearance and half-life elimination were shorter following IM administration (0.29 ± 0.05 mL/min and 4.27 ± 1.04 h) compared to IV administration (0.12± 0.01 mL/min and 7.94 ± 0.49 h), showing a similar pattern to that observed in undulate skates. The renal–portal system in elasmobranch fish could be responsible for the shorter Cl and t_1/2β_ after IM vs. IV administration, as blood from the caudal half of the body drains directly into the kidneys [44]. However, since both IM and IV administrations were performed in the caudal region of the animals, it remains unclear how elasmobranch physiology can affect drug elimination between different administration routes. In addition, these PK values suggest that an increase in the frequency of voriconazole administration could be needed to maintain voriconazole levels over the MIC50, and further studies using a multiple-dose regimen are needed to better understand this aspect.

The genes that are involved in drug metabolism can cause individual differences in response to medications. In humans, it is well established that voriconazole is metabolized in the liver by CYP450 enzymes, specifically CYP2C19, CYP2C9, and CYP3A4 [45]. Genetic variability in these enzymes can produce individual variations in drug metabolism, directly affecting plasma levels of voriconazole [46]. The metabolic pathways involved in the metabolism of azole antifungal drugs in fish are not well understood yet. A wide range of physiological variations between species can be involved in the different drug PK behaviors observed. Metabolic rate is related to activity level, ecology, cellular physiology, and kinematics, being very variable between species. It has been reported that in pelagic elasmobranch species, especially those with ram ventilation, the metabolic rate is higher when compared with benthic species [47]. As nursehound sharks and undulate skates are both benthic species, it is less likely that the variation between MRT and t_1/2β_ values is related to differences in the metabolic rate [28,48]. The osmoregulation and excretory system can also cause variations in drug metabolism and excretion. Elasmobranch osmoregulation is performed by the retention of nitrogen-derived metabolites (urea and trimethylamine oxide) by the kidneys and the elimination of NaCl through the gills and the anal gland (an exclusive organ in elasmobranchs) [47]. Despite that these characteristics could be related to variation in voriconazole elimination, insufficient data are available regarding how the unique excretory and osmoregulation system in elasmobranchs could be affecting voriconazole elimination.

In elasmobranchs, as well as in other species, the liver is responsible for basic metabolic functions such as the storage of nutrients, enzyme and hormone production, and the metabolism of xenobiotics [49]. One of the most important differences between the liver of teleosts and elasmobranchs is the lipid content, as in elasmobranchs the liver is the primary lipid storage organ [4]. In some animal models, a fatty liver is related to an inhibition of the activity of CYP450 enzymes, resulting in an alteration in PK parameters of certain drugs, decreasing the metabolism [50]. The size and composition of the liver in elasmobranchs vary between species and can account for up to 23% of their body weight, with the majority (up to 80%) potentially consisting of lipids. Furthermore, among elasmobranchs, the extent of hepatic cellular exposure, as well as the quantity and proportion of lipids, may influence the pharmacokinetic and pharmacodynamic properties of voriconazole [50,51].

Since most elasmobranchs are poikilothermic, their metabolism is closely tied to environmental conditions. Environmental factors such as temperature, salinity, and oxygen saturation have been shown to significantly influence the PK behavior of administered drugs across various fish species [4,47]. All these factors underscore the complexity of drug metabolism in aquatic organisms and the need for further research in this area.

Finally, potential adverse effects must also be considered, as azole antifungal drugs can cause hepatotoxicity, a side effect well-documented in human medicine but not fully understood in fish species [52]. It has been reported that hepatic toxicity in fish is less severe compared to mammals due to some anatomic and physiologic characteristics such as the ability of fish to tolerate extensive liver necrosis; ref. [53] suggests that these species could cope with higher doses of hepatotoxic drugs. Toxicity studies in teleosts have shown no adverse effects following the administration of sublethal doses of miconazole (25.22 mg/kg PO) to *Labeo rohita* [54]. However, it is well known that teleosts and elasmobranchs differ in liver function and composition. As lipid storage in the liver is fundamental for neutral buoyancy in elasmobranch species, it is possible that fat accumulation affects the susceptibility to liver toxicity of some xenobiotics in these species [4]. The potential toxic effect of azole antifungal drugs in elasmobranchs remains unknown, and there is a significant lack of information in this area. Despite no adverse effects being detected in our study, further pharmacodynamic, drug efficacy, and toxicity studies including higher doses and prolonged multiple-dose regimens are required to rule out or mitigate the potential adverse effects of azole drugs in elasmobranchs.

## 5. Conclusions

The results provided in this study suggest that the IM administration of voriconazole at a dose of 4 mg/kg in nursehound sharks produces faster absorption and disposition compared to undulate skates, reaching maximum plasma concentrations of 3.00 ± 0.23 µg/mL. The plasma concentrations achieved could be effective against *Fusarium solani* as they reach the MIC previously reported for these fungi (1–2 µg/mL). However, clinically relevant concentrations might be maintained for 6 h, which remains short. Due to the lack of information regarding the sensitivity of *Fusarium solani* in elasmobranchs, more evidence is needed before recommending the intramuscular administration of voriconazole in nursehound sharks for the treatment of fusariosis.

## Figures and Tables

**Figure 1 vetsci-12-00017-f001:**
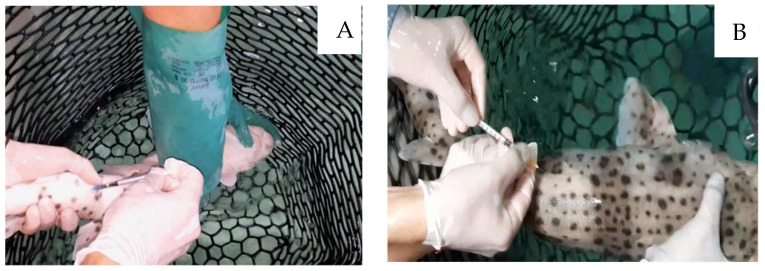
(**A**) Intravenous approach for drug administration and blood collection in a nursehound shark (*Scyliorhinnus stellaris*) during the PK study; (**B**) intramuscular drug administration during the PK study.

**Figure 2 vetsci-12-00017-f002:**
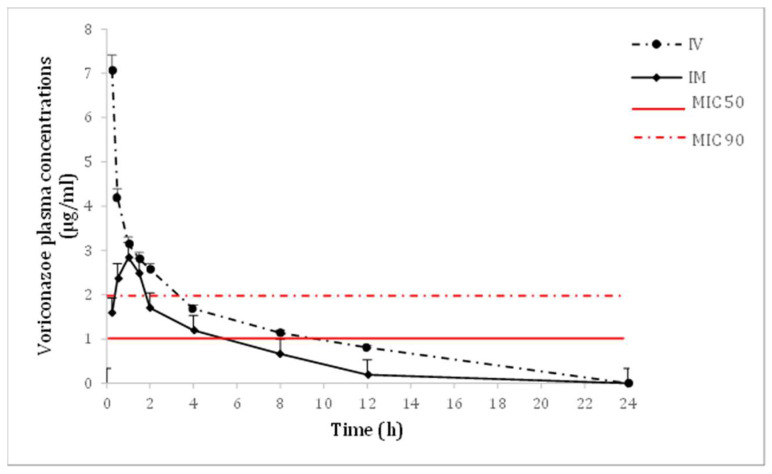
Mean ± SEM plasma concentrations of voriconazole (µg/mL) in nursehound sharks (*S. stellaris*; n = 6) under human care after administration of a single 4 mg/kg IV or IM dose. MIC50 and MIC90 values for *F. solani* [10].

**Table 1 vetsci-12-00017-t001:** Pharmacokinetic parameters of voriconazole administered at a single dose of 4 mg/kg by IV or IM route in nursehound sharks (*S. stellaris*; n = 6) and undulate skates (*R. undulata*) under human care as previously published [23].

	IV (n = 6)		IM (n = 6)		*p* (Wilcoxon Rank)	Undulate Skate	
PK Parameter (Unit)	MEAN	SEM	MEAN	SEM		IV	IM
C_0_ (μg/mL)	6.92	1.21	-	-	-	2.19	-
T_max_ (h)	-	-	1.08	0.15	-	-	1.33
C_max_ (μg/mL)	-	-	3.00	0.23	-	-	2.98
t_1/2β_ (h) *	7.94	0.49	4.27	1.04	0.026	11.18	9.59
AUC_t_ (μg/mL·h) *	24.33	3.22	10.06	1.72	0.004	47.46	27.92
AUC_inf_ (μg/mL·h)	33.63	3.12	15.91	2.92	0.078	58.14	37.60
MRT (h)	10.42	0.57	6.43	1.46	0.078	13.17	14.43
Vd (L/kg)	1.39	0.09	1.50	0.18	0.631	1.12	0.98
Cl (mL/min) *	0.12	0.01	0.29	0.05	0.006	0.07	0.12
F (%)	-	-	47.31	8.69	-	-	64.67
Tc > MIC50 (h)	10.00	-	6.00	-	-	18.00	10.00
Tc > MIC90 (h)	4.00	-	2.00	-	-	6.00	4.00

C_0_, extrapolated concentration at 0 h after IV administration; T_max_, time required to achieve maximum plasma concentration; C_max_, maximum plasma concentration; t_1/2β_, elimination half-life; AUC_t_, area under the plasma concentration curve computed using observed data points only; AUC_inf_, area under the curve extrapolated to infinity; MRT, mean residence time; Vd, distribution volume in pseudo-equilibrium conditions; Cl, clearance; F, bioavailability; Tc > MIC50, time that plasma concentration is above the MIC50 reported for *Fusarium solani*; Tc > MIC90, time that plasma concentration is above the MIC90 reported for *Fusarium solani* [10]. * Indicates the presence of statistically significant differences (*p* < 0.05 Wilcoxon signed-rank test) between IV and IM administration protocols.

## Data Availability

The data presented are contained within the article.

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
