# Peer review of "Voriconazole Pharmacokinetics Administered at 4 mg/kg IM and IV in Nursehound Sharks (Scyliorhinus stellaris) Under Human Care"

_vetsci, 2025, doi:10.3390/vetsci12010017_

Round 1
Reviewer 1 Report
Comments and Suggestions for Authors
This short communication is simple and rich in key information of PK sharks treated with Voriconazole via I.V and I.M , it should be considered as rapid or short communication for possible acceptance after minor revision
1, Line181 quantiication, error typing or not? check it?
2, Line 201-203, gramamr polishing for sentence? The mean recovery rate for voriconazole in nursehound shark plasma resulted 80 %, and was determined by adding know concentrations of voriconazole?
3, Line 392-393,reference 13, as: Exot. Anim. Form.; Carpenter, J.W., Marion, C.J., Eds.; Sixth edition.; Elsevier: St. Louis, Missouri, 2022; ISBN 9780-323-83392-9. , it should be corrected according to standard style?
4, Line 329, for Section 5 Conclusion, some key data or parameters as plasma concentration and MIC or other pharmecology/dynamic observation from Figure 1 and Table 1 should be presented
5, Finally, a dosage of 4 mg/kg voriconazole for two administration both I.V and I.M, why same or not different, any more reason or scentific explanation, yes, I see it was from your previous work/paper Ref 23
6, Any positive CK for this disease infected with same pathogen should be designed or cited together clearly fopr comparison at the same time? It is easy to do for you and it could be enhanced during Discussion if you have? 7, Line 50-51, Keywords I am not sure that eight words is suitable or permitted? So simple paper and so many key words? 8, Yes, I see some special English presentation was polished but more checking or polishment is necessary, perhaps? (END at 29 Nov 2024 at R119)Author Response
Please see the attachment

Reviewer 2 Report
Comments and Suggestions for Authors
The study presents a pharmacokinetic analysis of voriconazole administered IM and IV to Nursehound sharks, providing data on antifungal treatment in elasmobranchs. However, the limited dataset, inadequate study design, and restricted clinical applicability reduce its value as a full research article. Several key concerns require attention:
Plasma concentrations were discussed in relation to MIC values, but the clinical implications remain unclear. Specifically, it is uncertain how the PK findings translate into therapeutic outcomes or whether the brief duration above MIC values holds clinical significance. The small sample size (six sharks) further limits the generalizability of the findings. Additionally, the reported bioavailability (47.31%) is low. While the authors attribute this to leakage caused by skin and muscle characteristics, potential formulation improvements are not explored.
The manuscript briefly notes interspecies differences but lacks a detailed discussion of physiological factors influencing drug metabolism in elasmobranchs compared to other aquatic and terrestrial species. Moreover, the observed plasma concentrations did not exceed the reported MIC for Fusarium solani isolated from elasmobranchs (8 µg/ml), raising doubts about the dosing regimen’s effectiveness. Whether increasing the dose compromises safety and how this could be addressed?
In conclusion, given its limited data and outcomes, the manuscript is better suited as a short communication or brief report. Streamlining the introduction and discussion, while removing unnecessary elements (e.g., Figure 1 lacks scientific relevance), would enhance its clarity.
Reviewer 3 Report
Comments and Suggestions for Authors
This manuscript presents a comprehensive study on the pharmacokinetics of voriconazole administered intramuscularly (IM) and intravenously (IV) at a dose of 4 mg/kg in Nursehound sharks (Scyliorhinus stellaris). The study includes detailed methods for drug administration, blood sample collection, and pharmacokinetic analysis. The results provide valuable insights into the absorption, distribution, metabolism, and excretion of voriconazole in this species. The authors conclude that IM administration of voriconazole at 4 mg/kg produces faster absorption and disposition compared to IV administration, with plasma concentrations reaching levels that could be effective against Fusarium solani. Overall, it would be suitable for publication in Veterinary Sciences.
Specific Comments:
1. The study does not include any data on the potential adverse effects of voriconazole. Given the known hepatotoxicity of azole antifungals in other species, it would be beneficial to include a discussion on the safety profile of voriconazole in Nursehound sharks.
2. While the authors compare their findings with previous studies in other elasmobranch species, a more detailed comparison with pharmacokinetic data from mammals and other fish species could provide a broader context for the results.
3. Materials and Methods: The criteria for selecting the specific dose of 4 mg/kg should be more clearly justified, possibly referencing previous studies or guidelines for dosing in elasmobranchs.
Reviewer 4 Report
Comments and Suggestions for Authors
Could you please explain if other drugs such as Isavuconazole can also be used or what the comparison between Isavuconazole and voriconazole is in terms of effectivity?
Round 2
Reviewer 2 Report
Comments and Suggestions for Authors
Thanks to the authors for addressing my previous comments and providing responses, along with their arguments supported by additional references. I have included further suggestions that, if considered, could enhance the quality and informativeness of the manuscript:
- Table 1: Add explicit comparison columns between the observed PK profiles and those reported for other species (e.g., teleosts, reptiles, or even undulate skates, as referenced in a similar model study by the authors).
- Expand the discussion on why IM administration resulted in faster clearance and a shorter half-life compared to IV administration. Additionally, elaborate on how these findings could inform dosing regimens for treating fungal infections, particularly for achieving sustained plasma concentrations above MIC values.
